# Optimization of the Field Plate Design of a 1200 V p-GaN Power High-Electron-Mobility Transistor

**DOI:** 10.3390/mi13091554

**Published:** 2022-09-19

**Authors:** Chia-Hao Liu, Chong-Rong Huang, Hsiang-Chun Wang, Yi-Jie Kang, Hsien-Chin Chiu, Hsuan-Ling Kao, Kuo-Hsiung Chu, Hao-Chung Kuo, Chih-Tien Chen, Kuo-Jen Chang

**Affiliations:** 1Department of Electronics Engineering, Chang Gung University, Taoyuan 333, Taiwan; 2Department of Radiation Oncology, Chang Gung Memorial Hospital, Taoyuan 333, Taiwan; 3Department of Photonics, College of Electrical and Computer Engineering, National Yang Ming Chiao Tung University, Hsinchu 30010, Taiwan; 4National Chung-Shan Institute of Science and Technology, Materials and Electro-Optics Research Division, Taoyuan 333, Taiwan

**Keywords:** dynamic on-state resistance (R_on_), field plate (FP), high-temperature gate bias (HTGB), high-temperature reverse bias (HTRB), normally off operation, off-state breakdown voltage, p-GaN high-electron-mobility transistor (HEMT)

## Abstract

This study optimized the field plate (FP) design (i.e., the number and positions of FP layers) of p-GaN power high-electron-mobility transistors (HEMTs) on the basic of simulations conducted using the technology computer-aided design software of Silvaco. Devices with zero, two, and three FP layers were designed. The FP layers of the HEMTs dispersed the electric field between the gate and drain regions. The device with two FP layers exhibited a high off-state breakdown voltage of 1549 V because of the long distance between its first FP layer and the channel. The devices were subjected to high-temperature reverse bias and high-temperature gate bias measurements to examine their characteristics, which satisfied the reliability specifications of JEDEC.

## 1. Introduction

GaN power transistors have become key devices in high-power and high-efficiency power conversion systems because of their suitable material properties, such as a wide band gap, high electron mobility, and high critical breakdown field. Various approaches—such as using a gate-recessed structure [1,2,3], fluorine ion treatment [4], and adopting a p-type GaN cap layer—have been reported for enabling these devices to exhibit normally off operation [5,6,7]. Next-generation, high-power, and high-frequency switching systems are expected to be used in GaN power devices because of the superior characteristics of these systems. In particular, p-GaN power devices could be accepted in the market because the normally off operation offers a fail-safe requirement [8]. However, device reliability is a crucial factor that must be considered when GaN power devices are applied in high-power circuits. Many studies have examined the reliability of p-GaN high-electron-mobility transistors (HEMTs), using methods such as the high-thermal reverse bias test (HTRB) [9,10,11], high-temperature gate bias (HTGB)-stress-induced instability [12,13,14], hard-switching robustness [15], and short-circuit safe operating area [16,17,18]; however, further comprehensive research is still required on the long-term reliability of GaN HEMTs. The reliability of p-GaN HEMTs depends on their *V*_TH_ shifts [19], gate stress [20], and dynamic performance [21] as well as temperature [22].

In power switching applications, the devices are repetitively switched between ON and OFF states at high frequency. When the device is operated in OFF states, a high drain bias stress might affect the device breakdown voltage and *V*_TH_ shifts [23]. These shifts are caused by a strong electric field, electron trapping and detrapping, and hole deficiencies. This can be explained by the existence of a charge storage mechanism. When a high drain bias *V*_DSQ_ is applied to a p-GaN gate HEMT, the gate-to-drain capacitance (*C*_GD_) is charged to *Q*_GD_ (*V*_DSQ_). The net-positive charges at the drain-side access region are provided by the depletion of a two-dimensional electron gas. Therefore, drain-induced *V*_TH_ instability and large *V*_TH_ shifts are observed for p-GaN gate HEMTs when they are operated under a high drain voltage [24]. Another problem observed when p-GaN power devices are used for power switching is the dynamic on-state resistance (R_on_), which is measured immediately after the device is switched from the off state to the on state under a high drain stress. However, when a high drain stress is applied during the off state of the device, an electric field peak occurs at the gate edge, and some electrons might be accelerated and trapped in the surface or epitaxial layer of the device [25].

Some researchers have explored the failure mechanisms of normally off p-GaN HEMTs [26]. However, further research is required on this topic. According to Li, the *V*_TH_ shifts of a p-GaN gate HEMT under high-temperature reverse bias (HTRB) and NBTI stress occur because of hole emission in the p-GaN layer induced by the reverse bias. Li speculated that high temperatures can suppress the emission of holes or accelerate the detrapping of holes in experiments conducted under HTRB [27].

This study designed p-GaN power devices with different field plate (FP) setups (in terms of the number and positions of FP layers) and simulated the operation of these devices by using the technology computer-aided design (TCAD) software of Silvaco to examine their electric field between the gate and drain regions. The devices were then subjected to DC, off-state breakdown voltage, pulse, HTRB, and HTGB measurements.

## 2. Simulation Results

Three devices with different numbers of FP layers (zero, two, and three) located at different distances from the channel were designed. In this paper, the terms FP1, FP2, and FP3 denote the first, second, and third FP layers, respectively. The distributions of the electric fields of the devices under high drain bias were simulated using the TCAD software of Silvaco (Figure 1a–c). For the p-GaN HEMT without an FP (Figure 1a), most of the potential lines were concentrated around the drain side of the gate, which indicates that a high electric field peak can form for this device. For the p-GaN HEMTs with FP layers, high potential line density was observed at the drain side of the gate edge and the edges of FP1, FP2, and FP3. These results can be attributed to the modulation effect of the FPs, which disperse the electric field peak between the gate and drain regions.

## 3. Device Structure

p-GaN HEMTs were grown on 6-inch-diameter (15.24 cm-dia.) Si substrates through metal–organic chemical vapor deposition (MOCVD). A 300-nm-thick undoped GaN channel was deposited on top of a 4-μm-thick undoped GaN buffer transition layer. A 12-nm-thick Al_0_._17_Ga_0_._83_N layer, 1-nm-thick AlN etching stop layer, and 70-nm-thick p-GaN layer were then deposited in the channel. Finally, the wafer was annealed in an MOCVD chamber at 720 °C for 10 min in a N_2_ atmosphere, and the activated Mg concentration was determined to be 1 × 10^18^ cm^−3^ through Hall effect measurement.

First, p-GaN island etching was achieved through ICP with Cl_2_/BCl_3_/SF_6_, and the device was isolated through Ar^+^ ion implantation. Subsequently, a Ti/Al/Ni/Au layer was deposited as an ohmic metal and then annealed at 875 °C for 30 s in a N_2_ atmosphere. A SiN layer was then deposited on the device as a passivation layer, and a Mo/Au layer was stacked as the gate metal after the gate through etching. To disperse the electric field between the gate and drain regions, the first FP layer was deposited after the stacking of SiN. The second and third FP layers were then stacked on the device (Figure 2a,b). Table 1 presents the distances from FP1 to the gate, from FP1 to the channel, from FP2 to the gate, from FP2 to the channel, from FP3 to the gate, and from FP3 to the channel. The p-GaN power devices with FP layers had *L*_GS_, *L*_G_, *L*_GD_, and *W*_G_ values of 1.5 µm, 2 µm, 17.5 µm, and 100 mm, respectively.

## 4. Experimental Results and Discussion

The *I*_DS_–*V*_GS_, *I*_DS_–*V*_DS_, and *I*_GS_–*V*_GS_ characteristics of the devices with two and three FP layers are presented in Figure 3a–c, respectively. These characteristics were measured using the Agilent B1505A power device analyzer at room temperature, and the maximum current compliance was set to 20 A. No large differences were observed in the DC characteristics of the devices under a low gate bias and drain bias because the epitaxial structure of the devices was the same. Both devices exhibited a *V*_TH_ value of 2.2 V and an *I*_D__S_Leakage_ value of 1 × 10^−6^ A (Figure 3a). They exhibited a dynamic on-state resistance (R_on_) of 200 mΩ, an *I*_DS_saturation_ value of 15 A (@*V*_DS_ = 10 V, V_GS_ = 6 V), and an *I*_GS_Leakage_ value of 600 µA (@*V*_GS_ = 6 V) (Figure 3b,c). To demonstrate the function of the FPs, the off-state breakdown voltage was measured (Figure 3d). The devices with two and three FP layers exhibited off-state breakdown voltages of 1549 and 1353 V, respectively. Thus, both devices exhibited the same leakage level between 0 and 1000 V, and the device with two FP layers exhibited a lower leakage current than the device with three FP layers when the *V*_DS_ bias was higher than 1000 V. The highest off-state breakdown voltage can be attributed to the electric field in the vertical direction, which was highly dispersed at the gate edges. In general, FPs disperse the electric field at the gate edge effectively; thus, they can be used to improve the horizontal electric potential line. However, the vertical electric potential line is always ignored. As displayed in Figure 1b, when FP1 was closer to the channel, the potential line density between FP1 and the channel was higher. By contrast, when the distance between FP1 and the channel was longer, the potential line density between FP1 and the channel was sparser, which resulted in an increase in the device breakdown voltage under a high drain bias (Figure 1c).

For comparison with the international company, the measurement conditions were a pulse width of 10 µs and a period of 2000 µs. The time waveform of the device under a *V*_DSQ_ value of 600 V is displayed in Figure 4a. As displayed in Figure 4b, R_on_ was measured under different *V*_DSQ_ values from 0 to 600 V in 100 V increments. The R_on_ values of the devices with two and three FP layers were 1.36 and 1.31 times (@*V*_DSQ_ = 600 V), respectively. The R_on_ values of these devices increased as *V*_DSQ_ was increased from 0 to 400 V. As *V*_DSQ_ was further increased from 400 to 600 V, R_on_ values of the devices decreased. The aim of this study was to design an FP setup that can suppress the hot electron injection behavior at the gate edge under a high drain stress bias [28]. The results can be attributed to the electron trapping and detrapping as well as the two-dimensional hole gas (2DHG) in the devices when they were operated at different *V*_DSQ_ values. Therefore, the increase in R_on_ was caused by hot electron injection and the trapping and detrapping of electrons at the device surface or epitaxial interface during off-state operation, and then it could not be released when the device was turned on in the extremely short switching time. However, enhanced vertical leakage was observed in the entire area between the source and the drain [29,30], electron trapping was observed at donor defects [31], and the 2DHG formed at the interface of C-doped GaN. The formation of a strain relief layer [32] under a high *V*_DSQ_ value results in a decrease in the number of trapped electrons, which causes a reduction in R_on_ [33]. The energy band diagram for the substrate region under a high *V*_DSQ_ bias is displayed in Figure 4c.

The devices with two and three FP layers were subjected to HTRB and HTGB measurements. For commercial GaN power devices, the stress bias in HTRB measurement is set to 80% of the off-state breakdown voltage, and the stress bias in HTGB measurement is set to 100% of the gate turn-on voltage (reference from JEDEC). In this study, during the HTGB measurement, the drain and source were always grounded, the gate stress bias was 5 V, the chamber temperature was 150 °C, and the devices were under bias stress for 168 h. Figure 5a presents the time tracks of the gate leakage currents (@*V*_GS_ = 5 V) of both devices. The results indicate that the gate leakage currents of both devices were distributed between 200 and 600 µA, and the devices did not break down after 168 h under bias stress. During the HTRB measurement, the gate and source were always grounded, the drain stress bias was 600 V, the chamber temperature was 150 °C, and the devices were under bias stress for 168 h. Figure 5b displays time tracks of the drain leakage currents of both devices. The drain leakage currents of both devices were distributed between 50 and 200 µA, and the devices did not break down after 168 h under bias stress.

After HTGB measurement, the *I*_DS_–*V*_GS_, *I*_DS_–*V*_GS_, and *I*_GS_–*V*_GS_ characteristics of the devices with two and three FP layers were measured again to observe the changes in these parameters. After HTGB measurement, the device with two FP layers exhibited a *V*_TH_ shift of 15%, a 7% increase in R_on_, and a 15% increase in the gate leakage current. The device with three FP layers exhibited a *V*_TH_ shift of 14%, a 4% increase in R_on_, and a 6% increase in the gate leakage current. After HTRB measurement, the DC characteristics of the devices with two and three FP layers were remeasured to observe the changes in these characteristics. After HTRB measurement, the device with two FP layers exhibited a *V*_TH_ shift of 4%, a 4% increase in R_on_, and a 4% increase in the gate leakage current. The device with three FP layers exhibited a *V*_TH_ shift of 4%, a 4% increase in R_on_, and a 4% increase in the gate leakage current. The characteristics of the devices with two and three FP layers are presented in Table 2.

The results indicate that the device performance degraded after HTRB measurement. After subjecting a p-GaN HEMT to HTRB, surface defects and negative charges are generated, which can result in a negative potential. Consequently, the channel is depleted of electrons, and the gate depletion region is thus extended [34,35]. However, after a p-GaN HEMT is subjected to HTGB, a strong electric field can favor the generation of defects in p-GaN (similar to that observed in an AlGaN barrier under a negative bias), which leads to the generation of leakage paths and consequently the failure of the gate junction [36].

## 5. Conclusions

We compared the electrical characteristics of p-GaN HEMTs with different FP setups using the TCAD software of Silvaco. The DC characteristics of p-GaN HEMTs with two and three FP layers did not differ under low voltage operation. However, when these devices were operated under a high drain bias, the device with two FP layers exhibited a higher off-state breakdown voltage (1549 V) than the device with three FP layers (1353 V), which can be attributed to the higher dispersion of the vertical electric field for the device with two FP layers. Both devices exhibited low R_on_ values under high *V*_DSQ_ values because of the leakage current from the substrate region. The device with two FP layers had a lower manufacturing cost than that with three FP layers. Thus, the device with two FP layers was superior to that with three FP layers.

## Figures and Tables

**Figure 1 micromachines-13-01554-f001:**
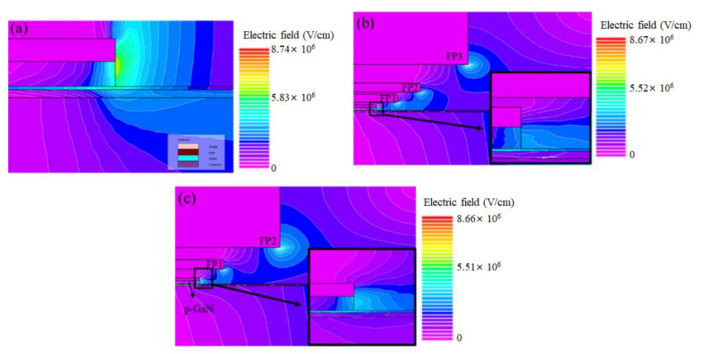
Electric field simulation results for a (**a**) p-GaN HEMT without an FP, (**b**) p-GaN HEMT with three FP layers, and (**c**) p-GaN HEMT with two FP layers.

**Figure 2 micromachines-13-01554-f002:**
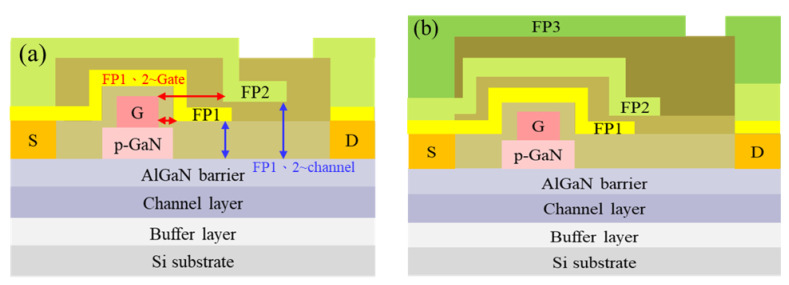
Cross-sectional schematics of the devices with (**a**) two and (**b**) three FP layers.

**Figure 3 micromachines-13-01554-f003:**
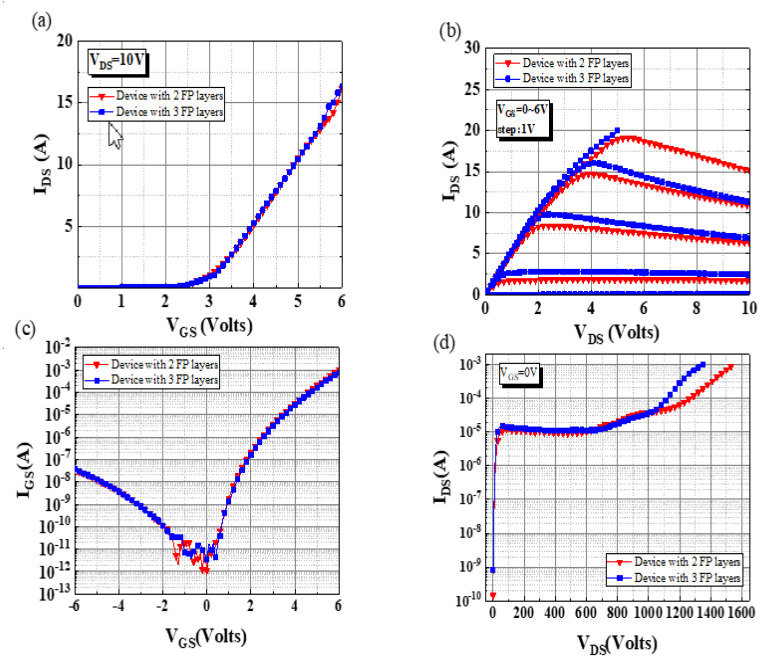
(**a**) *I*_DS_-*V*_GS_, (**b**) *I*_DS_-*V*_DS,_ (**c**) *I*_GS_-*V*_GS_, and (**d**) off-state breakdown voltage characteristics of the devices with two and three FP layers.

**Figure 4 micromachines-13-01554-f004:**
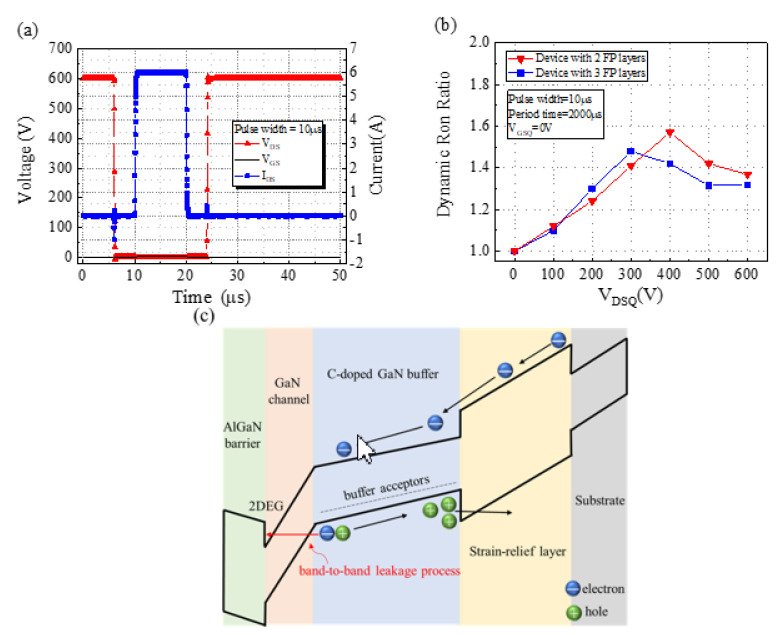
(**a**) Waveform of operating time during pulse measurement, (**b**) normalized R_on_ values under different *V*_DSQ_ values, and (**c**) energy band diagram for the substrate region under a high *V*_DSQ_ bias.

**Figure 5 micromachines-13-01554-f005:**
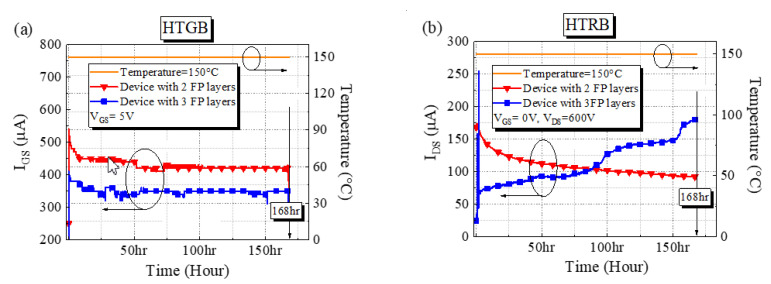
Time tracks of I_DS_ for the devices with two and three FP layers in the (**a**) HTGB measurement and (**b**) HTRB measurement.

**Table 1 micromachines-13-01554-t001:** Distances in the devices with two and three FP layers.

(Unit=μm)	* **L** * _ **SD** _	* **L** * _ **GD** _	*L* _GS_	FP1~Gate	FP1~Channel	FP2~Gate	FP2~Channel	FP3~Gate	FP3~Channel
Device with 2 FP layers	21	17.5	1.5	3	0.5	7	1.2	×	×
Device with 3 FP layers				2	0.3	4.75	0.7	10.5	1.7

**Table 2 micromachines-13-01554-t002:** Characteristics of the devices with two and three fp layers after the htrb and htgb measurements.

HTGB		*V* _TH_	R_on_	*I*_GS_@*V*_GS_ = 5 V
Temperature = 150 °C *V*_GS_ = 5 V Time = 168 h	Device with 2 FP layers	15%	7%	15%
	Device with 3 FP layers	14%	4%	6%
HTRB		*V* _TH_	R_on_	*I*_GS_@*V*_GS_ = 5 V
Temperature = 150 °C *V*_DS_ = 600 V, *V*_GS_ = 0 V Time = 168 h	Device with 2 FP layers	<4%	<4%	<4%
	Device with 3 FP layers	<4%	<4%	<4%

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
