# Peer review of "Optimization of the Field Plate Design of a 1200 V p-GaN Power High-Electron-Mobility Transistor"

_micromachines, 2022, doi:10.3390/mi13091554_

Round 1

Reviewer 1 Report

1200V p-GaN based power devices is promising for future applications. However, the reliability is still a crucial issue. In the present work, the peak electric field in the device is modulated by employing the field plate design, and finally the off-state breakdown voltage is increased. The results is well present. I think it can be published as it is.

Reviewer 2 Report

The author proposed and investigated the field plate (FP)  p-GaN power high–electron mobility transistors for high-power applications.

The content and novelty of the work are the strength of the paper.

I recommend this research article for publishing in Micromachines.

Reviewer 3 Report

This paper is recommended for Micromachines. The authors have done significant work. There are some suggestions in the paper:

Page 1, Line 31: The better terms are “high-electron mobility and high critical breakdown electric field.

Page 1, Line 36: A typo: normally-off operation

Page 2, Line 46: As the authors mentioned in the subsequent lines that the HEMT repetitively switches between ON and OFF states. I understand that there must be drain voltage variation during the power switching operation. However, in the current version, it is stated that “devices are operated between a high drain voltage and a low drain voltage”; seems like misleading. Please rewrite this sentence, it may confuse the reader.

Page 3, Line 102: There is an error “6-inch thick (15.24 cm thick) Si substrates”. Here, the numerical value indicates the diameter of the Si wafer (not thickness!), typically the Si substrates have the thickness about 300-400 µm. So, please correct as “6-inch diameter (15.24 cm dia.) Si substrates”

Page 3, Line 110: Ar+ ion-implantation

Page 3, Line 142: Agilent 1505A power device analyzer
